# The Impact of Tissue Preparation on Salivary Gland Tumors Investigated by Fourier-Transform Infrared Microspectroscopy

**DOI:** 10.3390/jcm12020569

**Published:** 2023-01-10

**Authors:** Mona Stefanakis, Miriam C. Bassler, Tobias R. Walczuch, Elena Gerhard-Hartmann, Almoatazbellah Youssef, Agmal Scherzad, Manuel Bernd Stöth, Edwin Ostertag, Rudolf Hagen, Maria R. Steinke, Stephan Hackenberg, Marc Brecht, Till Jasper Meyer

**Affiliations:** 1Process Analysis and Technology (PA&T), Reutlingen University, Alteburgstr. 150, 72762 Reutlingen, Germany; 2Institute of Physical and Theoretical Chemistry, University of Tübingen, Auf der Morgenstelle 18, 72076 Tübingen, Germany; 3Institute of Pathology, University of Würzburg, Josef-Schneider-Str. 2, 97080 Würzburg, Germany; 4Department of Oto-Rhino-Laryngology, Plastic, Aesthetic & Reconstructive Head and Neck Surgery, University Hospital Würzburg, Josef-Schneider-Str. 11, 97080 Würzburg, Germany; 5Chair of Tissue Engineering and Regenerative Medicine, University Hospital Würzburg, Röntgenring 11, 97070 Würzburg, Germany; 6Fraunhofer Institute for Silicate Research ISC, Röntgenring 11, 97070 Würzburg, Germany; 7Department of Otorhinolaryngology—Head and Neck Surgery, RWTH Aachen University Hospital, Pauwelsstr. 30, 52074 Aachen, Germany

**Keywords:** formalin, fixation, tissue preparation, salivary gland neoplasia, FTIR spectroscopy, principal component analysis, discriminant analysis

## Abstract

Due to the wide variety of benign and malignant salivary gland tumors, classification and malignant behavior determination based on histomorphological criteria can be difficult and sometimes impossible. Spectroscopical procedures can acquire molecular biological information without destroying the tissue within the measurement processes. Since several tissue preparation procedures exist, our study investigated the impact of these preparations on the chemical composition of healthy and tumorous salivary gland tissue by Fourier-transform infrared (FTIR) microspectroscopy. Sequential tissue cross-sections were prepared from native, formalin-fixed and formalin-fixed paraffin-embedded (FFPE) tissue and analyzed. The FFPE cross-sections were dewaxed and remeasured. By using principal component analysis (PCA) combined with a discriminant analysis (DA), robust models for the distinction of sample preparations were built individually for each parotid tissue type. As a result, the PCA-DA model evaluation showed a high similarity between native and formalin-fixed tissues based on their chemical composition. Thus, formalin-fixed tissues are highly representative of the native samples and facilitate a transfer from scientific laboratory analysis into the clinical routine due to their robust nature. Furthermore, the dewaxing of the cross-sections entails the loss of molecular information. Our study successfully demonstrated how FTIR microspectroscopy can be used as a powerful tool within existing clinical workflows.

## 1. Introduction

Salivary gland tumors (SGT) are responsible for approximately 3–6% of all head and neck neoplasms [1]. The considerable variety of benign and malignant SGTs becomes obvious by looking at the 2017 WHO Classification of Head and Neck Tumors, which differentiates between more than thirty different benign and malignant SGTs [2]. Unambiguous SGT classification and malignant behavior determination based on histomorphology alone can be challenging and, in some cases, impossible [3], which may complicate particularly a reliable intraoperative frozen section diagnosis.

As a consequence of digitalization in health care, digital pathology becomes more important within daily clinical workflows [4,5,6,7]. As part of this process, modern diagnostics could help to develop faster and more reliable tumor diagnostics, which are of great interest. Here, spectroscopic methodologies increasingly find their way to be used as effective diagnostic aids in tumor identification and demarcation. One such attractive method is Fourier-transform infrared (FTIR) microspectroscopy [8,9,10,11], which has already been introduced in various medical applications as a fast and robust technology [12,13]. This technique allows a spatially resolved sample analysis by detecting the different bonding vibrations of the tissue molecules. Commonly, FTIR spectroscopy is also combined with multivariate data analysis (MVA) to systematically reduce large data sets and extract the most relevant tissue-related information [14,15,16,17]. Many studies were performed using FTIR spectroscopy or imaging coupled with MVA in tumor diagnostics [18,19,20,21].

Although FTIR spectroscopic approaches are increasingly popular in tumor analysis, the already published studies focus on tumor identification. However, for assessing the translational potential of FTIR spectroscopy in routine diagnostics, knowledge about the influence on the tumor biology information of existing tissue fixation strategies is essential. The investigation of parotid SGTs is rare due to a lack of sample availability and quality. Some studies focused on the differentiation of SGT entities or their protein and lipid composition determination by other spectroscopic methods [3,22,23], but so far only few groups used FTIR spectroscopy for head and neck tumors in general [24].

Overall, the data of a spectroscopic analysis are directly related to the quality of the sample set, on which basis all measurements and evaluations are performed. The quality, however, directly correlates with the gained tissue itself and the sample preparation. As the preparations chemically affect the tissues, a good understanding of these changes is required. Typical tissue processing steps encompass cryoconservation, formalin fixation, paraffin or O.C.T.™ embedding and dewaxing [25,26,27,28,29]. However, only very few studies have addressed the effects of specimen preparation and fixation by spectroscopy methodologies [30,31,32,33]. Zohdi et al. analyzed the impact of formalin fixation on rodent heart and liver tissue sections by FTIR microspectroscopic analysis [34], and Hackett et al. investigated cryofixed and formalin-fixed murine brain tissue sections by FTIR imaging and PIXE elemental mapping [35]. Both studies recommend using freshly excised/cryofixed tissue. Recently, we investigated the effect of formalin fixation on human brain tumors and observed a high comparability to non-fixed tissue [36]. Besides the impact of tissue fixation on tissue constitution or morphology, the suitability of a fixation for a digital, spectroscopic-based diagnosis of tumors as well as its practical implementation into the clinical workflow are also of great importance.

In this study, we examined the influence of standardized preparation procedures on parotid salivary gland tissue, Warthin’s tumor and pleomorphic adenoma by FTIR microspectroscopy. For this purpose, four different preparation protocols were applied, including cryoconservation, formalin fixation, formalin fixation paraffin embedding (FFPE) and dewaxing. FTIR mean spectra of the differently prepared cross-sections and the used embedding or fixation chemicals are initially generated and evaluated. This comparison is performed for each of the three tissue types. Additionally, a chemometric approach is carried out by calculating a principal component analysis (PCA) combined with a Bayesian discriminant analysis (DA) in order to achieve an IR-spectra-based separation of all preparations. This allows us to reveal the spectral impact responsible for the preparation discrimination. The aim of this study is to ascertain the preparation method for parotid tissues, which has the least impact on the spectroscopic information. A high accordance between cryofixed and short-term formalin fixation can be observed. FTIR microspectroscopy can also be implemented within the clinical daily routine.

## 2. Materials and Methods

### 2.1. Patient Selection

Patients with a parotid gland tumor planned for parotidectomy were preoperatively screened and included in this study. Overall, 9 parotid gland tumors from 9 patients were collected, of which 5 tumors were identified by trained pathologists as Warthin tumors (age: 62.5 ± 10.7 years; male:female 2:3) and 4 tumors were diagnosed as pleomorphic adenomas (age: 55.9 ± 15.1 years; male:female 1:3). In the course of parotidectomy, surrounding areas of intact salivary gland tissue were unavoidably removed as well.

This study was approved by the institutional ethics committee on human research of the Julius-Maximilians-University Würzburg (vote 224/18). All experiments were performed according to the Declaration of Helsinki. All patients agreed to participate in this study by informed consent.

### 2.2. Preparation Workflow and Histologic Samples

Immediately after the parotidectomy, parotid tissue samples were sectioned, and one tissue piece containing tumor and adjacent salivary gland tissue each was chosen and prepared according to four different preservation protocols (hereafter described as “native”, “formalin”, “in paraffin”, “dewaxed”).

The fresh tissue specimen was frozen, sequential cross-sections were cut with a cryomicrotome (Leica CM 1900, Wetzlar, Germany) and stored at −80 °C. Based on this preparation, we defined these specimens as “native”. A second set of consecutive cross-sections was prepared identically, but was additionally fixed with 4% neutral buffered formalin (ROTI^®^Histofix, Carl Roth, Karlsruhe, Germany) incubated for 1 min. These samples were determined as “formalin”. After preparing the “native” and “formalin” cross-sections, the residual tissue piece was fixed overnight in 4% neutral buffered formalin and subsequently paraffin embedded (Thermo Fisher Scientific Inc., Waltham, MA, USA) using a routine automated approach. Here, the paraffin section preparation took place at ambient temperature. The paraffin-embedded tissues were stated as “in paraffin”. After the FTIR microspectroscopic analysis of the “in paraffin” sections, a dewaxing (ROTICLEAR^®^, Carl Roth, Karlsruhe, Germany) with subsequent rehydration steps was performed for each specimen. This sample set was declared as “dewaxed”. For each preparation method, 10 µm thick sequential sections were cut and placed on gold-coated slides (BioGold SuperChip, Thermo Fisher Scientific Inc., Waltham, MA, USA).

For histologic assessment, consecutive 3 µm thick cross-sections were cut immediately after sectioning the samples for each preparation method and stained with hematoxylin and eosin (HE) according to a standard protocol. Due to this approach, the spatial tissue composition can be directly transferred from the HE-stained sections to the spectroscopic cross-sections. HE-stained sections were placed on glass slides and digitized with a whole slide scanner (Hamamatsu Photonics, Hamamatsu, Japan) for diagnostic evaluation with the CaseViewer software (Version 2.4, 3DHISTECH, Budapest, Hungary). A simplified overview of the complete workflow is summarized in Figure 1.

### 2.3. Fourier-Transform Infrared Microspectroscopy

For each of the nine patients, four cross-sections according to the preservation protocols (“native”, “formalin”, “in paraffin”, “dewaxed”) were investigated. Three tumor and three salivary gland tissue regions were selected for each cross-section (Figure 2). Ten randomly chosen spectra were collected per tissue region resulting in 60 measurements of each tumor cross-section. In total, 2160 single FTIR spectra were recorded in reflectance mode with an infrared microscope (Autoimage, Perkin Elmer, Waltham, MA, USA) coupled to an FTIR spectrometer (FTIR System 2000, Perkin Elmer, Waltham, MA, USA) shown in Figure 1.

Following sample illumination, the reflected light was collected with a thermoelectric cooled mercury cadmium telluride (MCT) detector. The system was referenced against air and 256 accumulations for each measurement with a gain of 4 were acquired. The spectral resolution was 4 cm^−1^ and the optical path difference velocity was 2 cm/s. The wavenumbers range from 4000 cm^−1^ to 700 cm^−1^. An aperture size of 50 μm × 50 μm was used, which represents the measured area integrated for each single FTIR spectrum.

### 2.4. Data Pre-Treatment and Multivariate Data Analysis

Multivariate data analysis (MVA) was performed with The Unscrambler X 10.5 (Camo Analytics AS, Oslo, Norway). All spectra from 4000 cm^−1^ to 700 cm^−1^ were preprocessed in the same way: an area normalization followed by the 2nd Gap (smoothed) derivative (5 points). The wavenumbers from 4000 cm^−1^ to 3700 cm^−1^ and 2600 cm^−1^ to 1850 cm^−1^ were excluded due to variable reduction. The tissue types salivary gland tissue, pleomorphic adenoma and Warthin tumor were evaluated separately to investigate the effect of the preservation protocols “native”, “formalin”, “in paraffin” and “dewaxed”.

The PCA was calculated with mean centering, leverage correction and the singular value decomposition algorithm to distinguish between the tissue preparation protocols. Model outliers were identified in the influence plot Hotelling’s T^2^ vs. F-residuals (outlier limits 5% each). For the comparison of the tissue types, each PCA was combined with a Bayesian discriminant analysis (DA) with Mahalanobis distance (Warthin tumor and pleomorphic adenomas) or Euclidean distance (salivary gland tissue). The number of used principal components (PCs) for the DA was similar to the shown PCA models. The overall accuracy, sensitivity, specificity and precision were calculated based on the confusion matrix terminology (37, 38).

## 3. Results

### 3.1. Histologic Characterization of HE-Stained Samples

Pleomorphic adenoma and Warthin tumor are common benign SGTs and are macro- and microscopically well circumscribed. Histologically, pleomorphic adenomas show a mixture of ductal epithelial structures, myoepithelial cells as well as mesenchymal stromal elements in different proportions. In contrast, Warthin tumors are characterized by cysts and papillary structures lined by a two-layered oncocytic epithelium and associated lymphoid stroma with occasional lymphoid follicles with germinal centers (Figure 3).

### 3.2. Mean Spectra Analysis

In a first step, FTIR spectra were acquired to identify spectral changes resulting from each preservation procedure (“native”, “formalin”, “in paraffin”, “dewaxed”). FTIR mean spectra were calculated for the salivary gland tissue and both tumor entities using all measured single spectra (Figure 4). Additionally, FTIR spectra of the applied preservation and embedding chemicals (“formalin pure”, “paraffin pure”) were also acquired (Figure 4). Tissue-related and preparation-related influences can be deduced by comparing the spectral bands.

Figure 4 shows the FTIR mean spectra from 3700–700 cm^−1^ (2600–1850 cm^−1^ excluded) of the four preparation protocols (“native” (blue), “formalin” (red), “in paraffin” (grey), “dewaxed” (green)) as well as the used preparation chemicals (“formalin pure” (purple), “paraffin pure” (dark yellow)). Initially, a broad band maximum is noticeable at 3290 cm^−1^ for all preparations and tissue types (1), which can either be assigned to an O-H vibration from proteins or water (Table 1). This band, however, does not appear in the “pure paraffin” spectrum and is only slightly pronounced within the “formalin pure” spectrum (1, purple, yellow). Within a wavenumber region of 3000–2800 cm^−1^, a sum of three to four spectral bands is visible for the different preparations of all tissue types and the “paraffin pure” (2). These bands can be ascribed to combinations of CH_3_, CH_2_ vibrations, either resulting from lipids within the tissues or the paraffin itself (Table 1). Here, similar spectral trends occur between the “native” and “formalin” tissue as well as between the “in paraffin” tissue and the “paraffin pure”. On the contrary, the “dewaxed” tissue reveals a different spectral pattern in this region (2).

The mean spectra of the “native” and “formalin” tissue reveal a small spectral band at 1750 cm^−1^ for all tissue types, which is missing in all other preparation mean spectra (3). The 1750 cm^−1^ band is assignable to a C=O vibration of triglycerides or esters (Table 1). A sequence of four following FTIR maxima at 1660 cm^−1^ (4), 1544 cm^−1^ (5), 1466 cm^−1^ (6) and 1390 cm^−1^ (7) can be identified for all preparation protocols and tissue types as well as for the “paraffin pure” spectrum (6). These signals, however, do not occur in the “formalin pure” spectrum. All FTIR maxima in this wavenumber range can be deduced from C=O, amide I, N-H, amide II stretching vibrations and CH_3_, CH_2_ bending vibrations of either lipids or proteins (Table 1).

Additional FTIR bands appear within the fingerprint region of the four preparation procedures and tissue types. Two signals are particularly prominent at 1239 cm^−1^ and 722 cm^−1^. A distinct assignment of this spectral region is difficult, but maxima at 1239 cm^−1^ and 722 cm^−1^ (8, 9) can be matched with a PO^2−^, out-of-plane bending vibration of DNA/RNA and a stretching C-H vibration of lipids, respectively (Table 1). The latter band (9) is also highly dominant in the “in paraffin” tissues. FTIR spectra of the preparation chemicals “formalin pure” and “paraffin pure”, however, do not show a pronounced band pattern within the fingerprint region.

A summary of all identified IR bands is listed in Table 1. Although no distinct assignment of bands within the fingerprint region is possible, this wavenumber range is very characteristic.

### 3.3. FTIR Data Analysis by PCA-DA

Statistical analysis by PCA-DA enables to reveal the smallest spectral influences and thus provides the basis to discriminate the differently prepared parotid tissues. Figure 5 shows the PCA models for all three tissue entities (salivary gland tissue (Figure 5a), Warthin’s tumor (Figure 5b) and pleomorphic adenoma (Figure 5c)), distinguishing the four preparation protocols (“native” (light blue), “formalin” (green), “in paraffin” (red), “dewaxed” (blue)). Spectral ranges of 3700 cm^−1^ to 2600 cm^−1^ and 1850 cm^−1^ to 700 cm^−1^ were considered for calculating the models. The 3D scores plot of the salivary gland tissue reveals a separation into three groups (a1). Three PCs are required to explain 80% of the data (PC1 54%, PC2 17% and PC3 9%). PC1 demarcates the below-average “in paraffin” group from the other clusters, which are aligned above-average. PC2 arranges the “dewaxed” group as below-average, the “in paraffin” group as in-average and the “formalin” and “native” group as above-average. PC3, however, exhibits only minor differences between the “native” and “formalin” group. Reasons for this clustering are displayed in the corresponding loadings plots of PC1, PC2 and PC3 (a2). All three loading plots show a major impact in the wavenumber region of 3000–2800 cm^−1^, 1700–1500 cm^−1^ and the fingerprint region below 1500 cm^−1^. The loadings plot of PC1 distinguishes three main bands in a wavenumber range of 3040–2800 cm^−1^, 1500–1440 cm^−1^ and 740–700 cm^−1^. In the PC2 loadings, an overall higher spectral influence is noticeable within wavenumber ranges of 3040–2810 cm^−1^, 1780–1610 cm^−1^, 1500–1360 cm^−1^, 1160–1050 cm^−1^ and 750–700 cm^−1^. Similar spectral effects are shown in the PC3 loadings (3010–2820 cm^−1^, 1790–1700 cm^−1^, 1500–1360 cm^−1^, 1220–1090 cm^−1^, 750–700 cm^−1^).

The 3D scores plot of the Warthin’s tumor also demonstrates a segregation into three groups according to the different preparation procedures. A total explained variance of 84% was reached by the first three PCs (PC1 65%, PC2 15% and PC3 4%). For PC1, the “in paraffin” group is separated from the below-average groups “formalin”, “dewaxed” and “native” due to its above-average positioning. PC2 reveals an above-average arrangement of the “native” and “formalin” group, whereas the “dewaxed” one is positioned below-average. The “in paraffin” group lies in-between both and is thus organized as in-average. PC3 additionally displays an in-average arrangement of the “in paraffin” and “dewaxed” group. “Formalin” and “native” prepared groups, however, are either positioned above- or below-average by tendency. This overall separation results from the spectral information represented by the corresponding loading plots of all three PCs (b2). The PC1 loadings reveal three distinct wavenumber regions (3000–2810 cm^−1^, 1500–1350 cm^−1^, 750–700 cm^−1^), which mostly affect the group separation. Compared to the PC1 loadings, the PC2 loadings plot exhibits a more widespread spectral impact, including wavenumber ranges of 3700–3550 cm^−1^, 3020–2800 cm^−1^, 1775–1580 cm^−1^, 1500–1400 cm^−1^, 750–700 cm^−1^. A comparable spectral influence is also discernible in the PC3 loadings plot with two additional maxima at 1750 cm^−1^ and 1470 cm^−1^.

The 3D scores plot of the pleomorphic adenoma shows a similar group separation for all investigated preparation approaches (PC1 84%, PC2 4% and PC3 2%, explained variance 90%). As described earlier for the Warthin’s tumor, the above-average “in paraffin” group is completely separated from the below-average groups “native”, “formalin” and “dewaxed” by means of PC1. The demarcation due to PC2, however, reveals a below-average positioning for the “dewaxed” group, an above-average arrangement of the “formalin” and “native” group and the “in paraffin” group is in-average. For PC3, the “dewaxed” and “in paraffin” groups are represented as in-average position whereas the “formalin” group is tendentially organized above-average and the “native” one tendentially ranges below-average with regard to PC3. The arrangement of the clusters in the 3D scores plot is identical to the Warthin’s tumor evaluation. The reasons for this segregation can be deduced from the corresponding loading plots (c2). The PC1 loadings reveal three wavenumber regions (3000–2810 cm^−1^, 1500–1430 cm^−1^, 750–700 cm^−1^). Comparable wavenumber ranges are also identified in the PC2 loadings with additional impacting wavenumber regions of 3700–3550 cm^−1^ and 1770–1370 cm^−1^ as well as a dominating maximum at 1470 cm^−1^. The spectral influence in the PC3 loadings is also very similar to the range described by PC2 loadings, except for two additional maxima at 1750 cm^−1^ and 1470 cm^−1^ as well as the wavenumber range between 1230–950 cm^−1^.

In all three models, PC1 distinguishes “in paraffin”, whereas PC2 differentiates “dewaxed” and PC3 is not possible to clearly demarcate between the “native” and “formalin” preparation procedure. Therefore, the PCA scores were used in a DA to quantitatively calculate the clustering. This is based on the confusion matrix values. Confusion matrices of each model are listed in the Appendix A. An overview of the PCA-DA model parameters is summarized in Table 2.

Due to an overlap of the “native” and “formalin” group within the PCA, the misclassification of “formalin” and “native” is distinct (e.g., 40% native pleomorphic adenoma), whereas in the “in paraffin” and “dewaxed” group almost all spectra are predicted correctly (e.g., 100% in paraffin salivary gland tissue). This is attributed to the similarity between “native” and “formalin”. All three tissue types exhibit model parameters higher 75% and thus built robust models. If “native” and “formalin” are combined as one class and a PCA-DA is performed, the model quality parameters reached over 97%. Confusion matrices of each model are listed in the Appendix A. An overview of the PCA-DA model parameters is summarized in Appendix A.

## 4. Discussion

Tissue evaluation and diagnosis in frozen as well as FFPE sections is based on histomorphological criteria after HE staining. However, in many situations, additional examinations, currently in particular immunohistochemical stainings, are an important aid in establishing the correct diagnosis. FITR spectroscopy gives information about the molecular composition of tissues, without destroying the tissue within the measurement process. Tissue preparation and fixation for preservation and subsequent analysis are crucial steps in clinicopathological routines to establish a qualified diagnosis. In histopathology routine diagnostics, mainly cryo-fixed and FFPE dewaxed tissue preparation is performed. For this reason, the impact of common tissue fixation methods and their derivatives needs to be understood in order to deduce their applicability for spectroscopic analysis as diagnostic tool. Thus, preparation-specific influence on the spectroscopic bioinformation in salivary gland tissue, Warthin tumor and pleomorphic adenoma were investigated by FTIR microspectroscopy.

By comparing FTIR mean spectra of differently prepared parotid tissues among each other and with the fixation and embedding chemicals, spectral band variations and similarities can be identified (Figure 4). In a wavenumber range of 3000–2800 cm^−1^ (2) (Figure 4, Table 1), the “dewaxed” spectrum reveals a different IR signature compared to the other preparations and the “paraffin pure” spectrum. We expect these differences to result from tissue alterations due to dewaxing, which directly affects the tissue’s nature. After tissue dewaxing especially an extraction of fat factions is expected. Additionally, spectral bands of “native” and “formalin” are highly similar at 3000–2800 cm^−1^, which leads to the conclusion that the formalin fixation does not have an impact on these bands. Moreover, signal intensities within 3000–2800 cm^−1^ are highest for the “in paraffin” and lowest for the “dewaxed” mean spectra compared to the other preparations (Appendix A). This indicates that the incorporated paraffin contributes to the CH_3_/CH_2_ vibrations within the “in paraffin” spectrum. The low signal intensity in “dewaxed”, however, suggests that not only paraffin, but also proteins or lipids were removed from the tissues by dewaxing. Furthermore, the IR band at 1466 cm^−1^ (6) can be detected in all preparations, except for the “formalin pure” spectrum (Figure 4, Table 1). Its highest signal intensity is observed in the “paraffin pure” and the “in paraffin” spectra compared to the other preparations. This shows again that the “in paraffin” signal intensity is amplified by the embedded paraffin, as was already observed for 3000–2800 cm^−1^. In the fingerprint region, the band pattern differ between “native”/“formalin” and “in paraffin”/“dewaxed”, indicating that already paraffin and also the dewaxing affected the molecular composition. The results are consistent with formalin having only a minor effect on the molecular composition due to its crosslinking mechanism.

A PCA was calculated to identify the effects of each preparation method, since only minor changes in terms of band signature, shape and intensity are observed in the spectra. A summary of the PCA analysis for the analyzed preparation methods is illustrated in Figure 5. The 3D scores plots of all tissue types reveal a similar group arrangement for the “in paraffin”, “dewaxed”, “formalin” and “native” preparations (Figure 5(a1–c1)). PC1 distinctly separates the “in paraffin” from the other treatment groups. Thus, the greatest influence is assumed to be derived by paraffin. Three specific spectral areas can be addressed, which affect the “in paraffin” positioning (3000–2800 cm^−1^, 1466 cm^−1^, 722 cm^−1^) (Figure 5(a2–c2)). These can be correlated to CH_2_/CH_3_ and C-H signals from paraffin, but also to vibrations of tissue components. Therefore, these band regions can be interpreted as a paraffin overlap on top of the CH_2_/CH_3_ and C-H signals of the tissues. PC2, however, clearly distinguishes between the “dewaxed” and “native”/“formalin” group in all 3D scores plots. By comparing the PC2 loadings of all tissue types (Figure 5(a2–c2)), an increased impact of the C=O and N-H bonding vibrations (1750–1500 cm^−1^) is observed, whereas a decrease in CH_2_/CH_3_ and C-H maxima (3000–2800 cm^−1^) is noticed. Additionally, the fingerprint region is also more pronounced. The influence of 1750–1500 cm^−1^ and the fingerprint region (1050–500 cm^−1^) can be ascribed to a C=O band at 1750 cm^−1^ and a weak IR signal at 722 cm^−1^, which are only present in the “native”/“formalin” spectra, but are missing in the “dewaxed” spectrum. Thus, both signals mainly contribute to the separation of “dewaxed” from “native”/“formalin” on PC2. An explanation is again that dewaxing not only extracts paraffin, but also other lipophilic substances from the tissues and consequently this spectral information is removed from the “dewaxed” spectrum. Furthermore, a pronounced influence of the 1750–1500 cm^−1^ area is illustrated for the salivary gland tissue in the PC2 loadings. This can be explained by the significantly higher amount of lipid vacuoles and adipocytes in salivary gland tissue compared to both tumor entities. Therefore, this 1750–1500 cm^−1^ region represents mainly the influence of lipids for the salivary gland. PC3 additionally indicates a partial distinction of the “native” and “formalin” group for all tissue types. The associated PC3 loading plots (Figure 5(a2–c2)) show the highest discrimination influence again at lipid- and protein-corresponding wavenumbers (2855 cm^−1^ CH_2_/CH_3_, 1750 cm^−1^ C=O, 1134 cm^−1^ CH_2_/CH_3_). One explanation is that lipids, in comparison to proteins, might not be preserved by formalin fixation, which causes a partial differentiation. This is supported by various studies, which state that formalin is only partially or even not able to preserve lipids [39]. However, others claim formalin to be appropriate for this purpose [40]. Another reason for the high overlap between the “native” and “formalin” groups is that the formalin fixation on the one hand stabilizes the molecules by their cross-linking and on the other hand the native tissue degradation only slowly proceeds, so that tissue compositions are assumed to be almost identical for both. Additionally, an increasing influence of 1750–1500 cm^−1^ and 1500–1050 cm^−1^ is shown for the tumor entities in the PC3 loadings. These might point to a tissue heterogeneity, of which pleomorphic adenoma is the most heterogeneous one, followed by the salivary gland tissue and the Warthin tumor.

All findings are further confirmed by the additionally performed DA (Table 2). The PCA-DA shows the best classification results for the “in paraffin” and “dewaxed” groups with correct prediction outcomes between 95–100% of all tissue types. Compared to that, the “formalin” group is 70–87% less accurately predicted. The “native” group, however, is classified even worse with 64% for salivary gland tissue, 69% for Warthin tumor and 40% for pleomorphic adenoma. These classifications show that despite the pronounced group overlap between “native” and “formalin”, 70–87% of the “formalin” group were still correctly identified. This demonstrates the high stabilization effects of the tissues by the formalin method and thus its robustness. The poor classification for the “native” group is influenced by the tissue homogeneity, as the very heterogeneous pleomorphic adenoma is subjected to the worst prediction results, whereas the most homogeneous Warthin tumor achieved the best outcome in this group. This observation is well allegeable because a high tissue heterogeneity results in more variant spectroscopical results. Overcoming the influence of tissue heterogeneity on the validity of prediction models could be challenging to address in the future. Nevertheless, overall accuracy, sensitivity, specificity and precision are above 75% for all tissue types, demonstrating the models to be very robust. If “native” and “formalin” are grouped as one class due to their similarities, all model quality parameters are above 97%. However, the aim of this study was not to investigate the entity assignment of salivary gland tumors. The chosen model entities Warthin tumor and pleomorphic adenoma are easy to distinguish from each other based on classical histomorphology criteria.

An accurate patient diagnosis is directly linked to the quality and information content gained from the tissue samples, which are also influenced by the respective tissue preparation protocol. Therefore, the effects of these preparations on tissues and the spectral bioinformation are important to know. As demonstrated by our FTIR investigations, the different preparation methods have an impact on the chemical composition of the parotid tissues. The paraffin embedding and dewaxing were shown to either superimpose the signals or cause a loss of information due to additional tissue component removal. For native and short-term formalin-fixed tissues, the chemical structure is very similar and thus a differentiation based on their chemical information is almost not possible. Based on our results, short-term formalin fixation is declared to be the most appropriate preparation method, not only from a highly stabilizing and less interfering conservation aspect, but also from its suitability for spectroscopic tissue analysis. In histopathology routine diagnostics, mainly cryo-fixed and FFPE dewaxed tissue preparation is performed. A big advantage of FFPE dewaxed tissues is that many samples, also of rare tumor entities, are available for building spectroscopy training data sets. However, the fixation process of FFPE dewaxed samples takes a while, therefore it is not suitable for frozen sections. From a spectroscopic point of view, FFPE and native tissues could also be used for a molecular assessment by FTIR with PCA-DA, but the risk of tissue-related artefacts or degradation while measuring is increased. In comparison to other spectroscopic techniques, FTIR is a fast, reliable method, which measures the chemical information of a sample by exciting IR-active transitions. In particular, the fingerprint region of FTIR spectra is very characteristic for each sample and often allows a clear identification. This is a great advantage for investigating preparation-associated differences of parotid tissues. In contrast, Raman or fluorescence spectroscopy are either time-consuming acquisition techniques or require the application of fluorescence markers, which hampers the overall data measurement or might cause interferences by applying additional chemicals. As a result, these methods might be less appropriate for the purpose of this study. As spectroscopic diagnostics are more and more implemented in clinics, a suitable tissue preparation is mandatory. Additionally, formalin fixation allows an easy handling and thus good integration in a daily clinical workflow for the preservation of salivary gland tissue and tumors.

## 5. Conclusions

Since the effects of tissue preparation methods are hardly explored, a better understanding is required, especially in the context of digital diagnostics with optical spectroscopy. This is of high importance also in a further development of parotid tumor diagnosis. For this purpose, the impact of four classical sample preparations on salivary gland and parotid tumor tissues was investigated by FTIR microspectroscopy. Spectral variations among FTIR mean spectra of the differently prepared tissues and the used preservation chemicals were initially identified. Main differences were recognized in a wavenumber region of 3000–2800 cm^−1^, in the fingerprint region and for a spectral band at 1466 cm^−1^. Pattern variations between 3000–2800 cm^−1^ were mainly ascribed to effects of paraffin or dewaxing, indicated by either a paraffin-related enhancement of CH_2_/CH_3_ oscillations or a dewaxing-associated removal of lipids or proteins. Additionally, the 1466 cm^−1^ band also represents the amplifying influence of paraffin on CH_2_/CH_3_ signals, whereas the fingerprint region mostly differentiates between “native”/“formalin” and “in paraffin”/“dewaxed”. This already implied the impact of the different preparation methods on the chemical nature of the tissues. No spectral discrimination of “native” and “formalin” was achieved by FTIR mean spectra comparison and thus no influence on the chemical composition of the tissues by formalin fixation was ascertained. Due to only minor IR differences, a subsequent PCA-DA was performed to determine the spectral effects caused by sample preparation and to deduce its impact on the chemical composition of parotid tissues. A complete PCA separation according to the preparations was achieved for the “in paraffin” and “dewaxed” groups. Our results revealed that the embedded paraffin often overlaid or amplified the CH_2_/CH_3_ vibrational signals of the tissues. Furthermore, tissue dewaxing caused (as expected) not only paraffin removal, but also partial removal of tissue components, especially lipids, and thus a loss of information. The “formalin” and “native” group mainly superimposed in PCA score plots due to the high chemical similarity, also reflected by the comparable IR patterns. We assumed the stabilizing effects by formalin and the just slowly proceeding decomposition of native tissue to be responsible for their high overlap. All results were also reflected by the calculated quality parameters for the four classes PCA-DA model above 75%. In case, native and formalin-fixed tissues are combined to one class, in the resulting three classes the PCA-DA model reached quality parameters above 97%. This confirms the high chemical similarity between native and formalin-fixed tissue. Due to the overall stabilizing and low interfering impact of the short-term formalin fixation, its usage is highly suitable for parotid tissue treatment, especially in the context of a spectroscopic-based diagnosis. Additionally, formalin fixation is easily implementable in a tissue preparation workflow within the clinical daily routine, facilitating the transfer from science to application. In further studies, the stability of short-term formalin-fixed tissue at room temperature will be investigated. To conclude, FTIR spectroscopy provides additional information that may aid in a reliable tumor characterization, but homogenous tissue pretreatment processes are essential for comparability of such measurements for possible future diagnostic (routine) application.

## Figures and Tables

**Figure 1 jcm-12-00569-f001:**
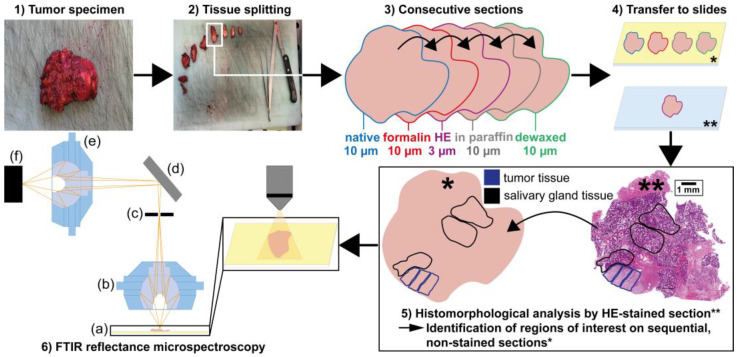
Workflow of tissue preparation for FTIR microspectroscopic measurements. Salivary gland and tumor tissues were resected (**1**) and cut into pieces (**2**). One tissue piece was chosen and prepared according to four different preparation protocols (“native”, “formalin”, “in paraffin” and “dewaxed”) and complemented with a separate HE staining (**3**). Cross-sections were prepared and mounted onto gold-coated objective slides for the four different preparation protocols for microspectroscopy (**4***) and onto glass slides for the HE staining and conventional microscopic analysis (**4****). The asterisks * is used to highlight the gold-coated objective slides, ** for the glass slides. The HE-stained sections were used to identify regions of interest on sequential non-stained sections (**5**). Finally, the “native”, “formalin”, “in paraffin” and “dewaxed” cross-sections were measured in reflectance mode with a FTIR microspectrometer (**6**): in (**a**) tissue sample on a gold-coated slide, (**b**) Cassegrain objective, (**c**) remote aperture position, (**d**) mirror for excitation and detection light path, (**e**) Cassegrain MCT, (**f**) MCT detector.

**Figure 2 jcm-12-00569-f002:**
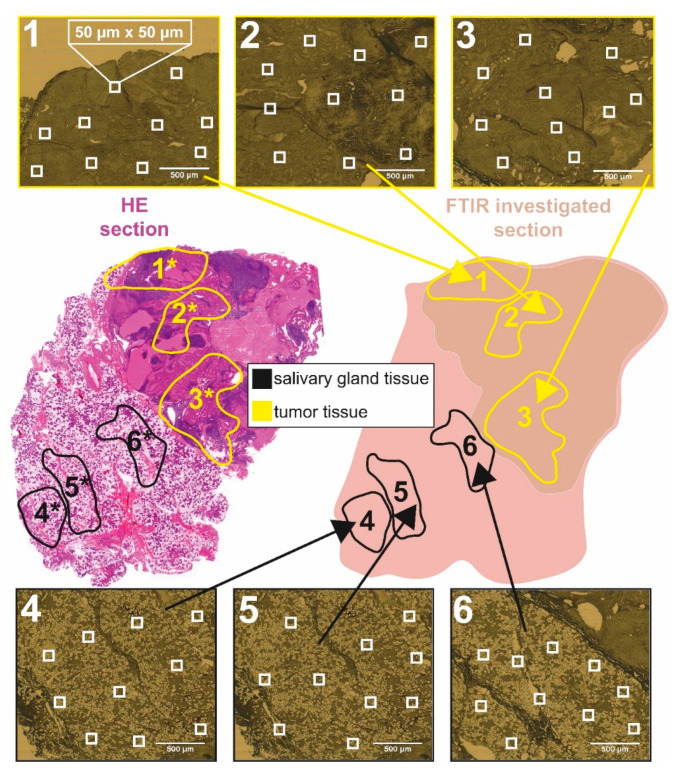
Identification of different tissue measurement regions for FTIR microspectroscopy. For each preservation method, standardized HE-stained sections were used to assign various tissue regions of interest, defined as regions **1***–**6*** (HE-section left). Identical salivary gland and tumor regions were defined in consecutive, non-stained tissue sections, schematically illustrated in region **1**–**6** (schematic section right). These predefined regions were additionally captured by the FTIR video image camera (images **1**–**6** on top and bottom). Within the video images, 10 single FTIR spectra were randomly acquired in a measuring area of 50 µm × 50 µm (white squares).

**Figure 3 jcm-12-00569-f003:**
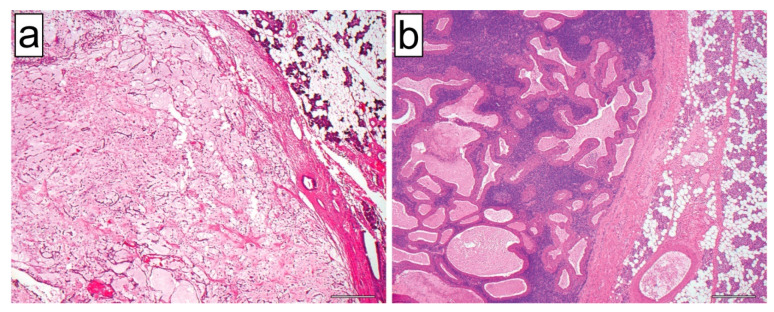
HE-stained examples of a pleomorphic adenoma (**a**) and a Warthin tumor (**b**) examined in this study (original magnification 40×; length of the scale bar: 50 μm). In both images, tumor-free salivary gland parenchyma is included on the right side of the image and a sharp demarcation of the benign tumor from the surrounding parenchyma can be seen.

**Figure 4 jcm-12-00569-f004:**
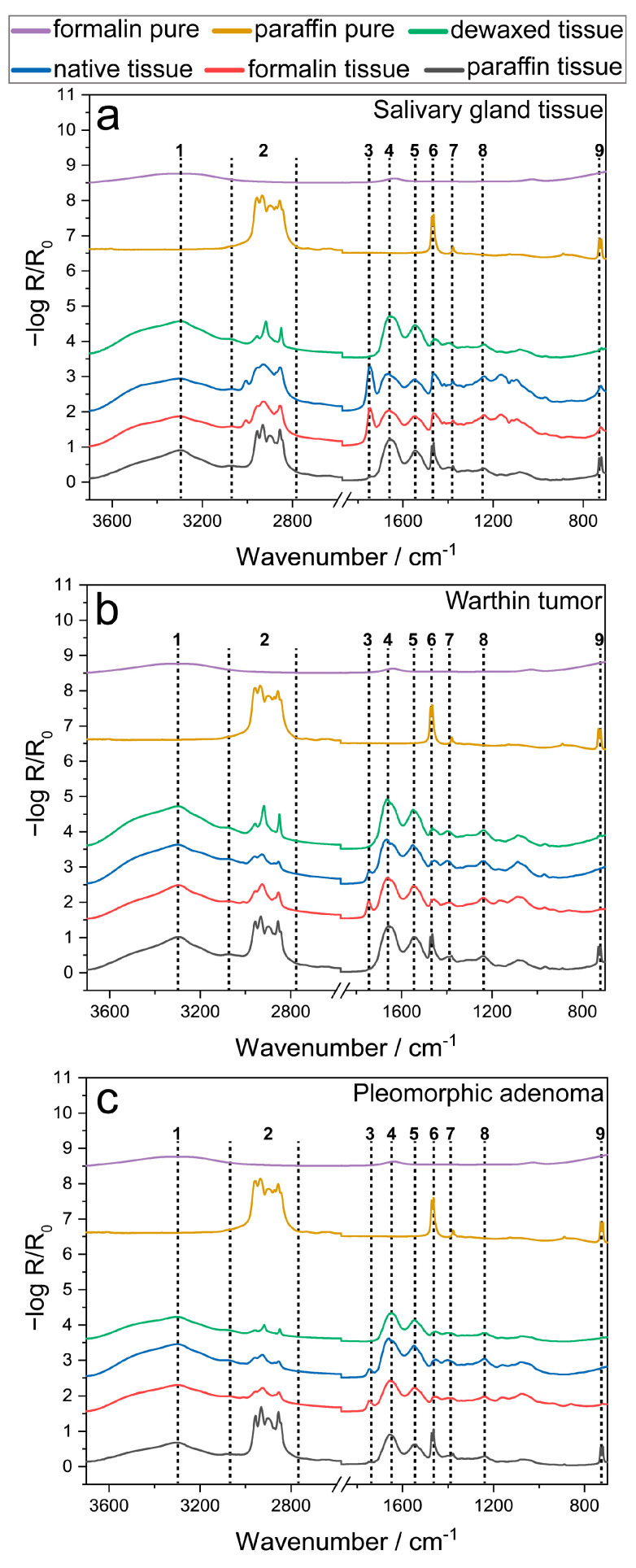
FTIR average spectra of each preservation method (“native” (blue), “formalin” (red), “in paraffin” (grey), “dewaxed” (green)) for the three investigated tissue types, salivary gland tissue (**a**), Warthin tumor (**b**) and pleomorphic adenoma (**c**). “Formalin pure” (purple) and “paraffin pure” (yellow) average spectra were additionally presented to reveal the IR oscillations resulting from the preparation chemicals. All average spectra are placed by an offset for an improved comparability of signals, also indicated by the dotted lines and numbers (1–9).

**Figure 5 jcm-12-00569-f005:**
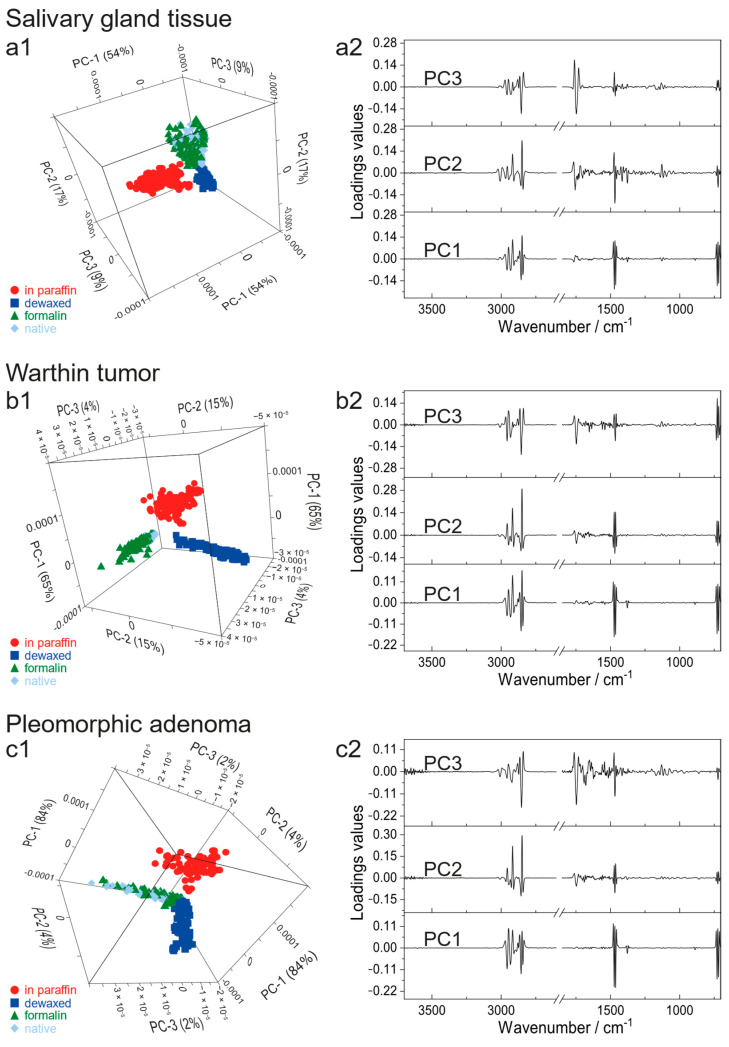
PCA models for the salivary gland tissue (**a**), Warthin’s tumor (**b**) and pleomorphic adenoma (**c**) in terms of the different preservation procedures (“native” (light blue), “formalin” (green), “in paraffin” (red), “dewaxed” (blue)). (**a1**,**b1**,**c1**) represent the 3D scores plots, whereas (**a2**,**b2**,**c2**) show the corresponding loadings.

**Table 1 jcm-12-00569-t001:** Spectral band assignment for the salivary gland tissue, Warthin tumor, pleomorphic adenoma, formalin pure and paraffin pure FTIR average spectra. All assignments are mainly based on [37,38].

Number	Wavenumber/cm^−1^	Assignment	Cause
1	3290	*ν* N-H amide, *ν* O-H	proteins, water
2	3000–2800	*ν* + *ν*_as_ CH_3_, CH_2_	paraffin, lipids
3	1750; 1700–1500	*ν* C=O triglyceride and ester, protein region	lipids
4	1660	*ν* C=O, amide I	proteins
5	1544	*δ* N-H, amide II	proteins
6	1466	*δ* + *δ*_as_ CH_3_, CH_2_	lipids
7	1390; 1350–1000	*δ* + *δ*_as_ CH_3_, CH_2_, stretching vibration CH_2_	lipids
8	1240/1239; 1000–700	*ν* + *ν*_as_ PO_2_^−^, out of plane bending vibration	DNA/RNA
9	722	*ν* C-H	lipids

**Table 2 jcm-12-00569-t002:** Model quality parameters for the salivary gland tissue, Warthin tumor and pleomorphic adenoma PCA-DA models. The robustness of the models can be deduced from the total and percentage amount of correctly assigned FTIR spectra for each preservation method. Based on this assignment, the average model parameters accuracy, sensitivity, specificity and precision can be calculated.

Entity	Fixation Method	Total Spectra	Correctly Predicted	Correctly Predicted /%	Accuracy /%	Sensitivity /%	Specificity /%	Precision /%
salivary gland tissue	in paraffin	270	270	100	83	84	94	83
	formalin	270	192	71				
	native	270	174	64				
	dewaxed	269	260	97				
Warthin tumor	in paraffin	150	150	100	89	89	96	90
	formalin	150	131	87				
	native	150	103	69				
	dewaxed	150	150	100				
pleomorphic adenoma	in paraffin	120	120	100	76	77	92	76
	formalin	120	84	70				
	native	120	48	40				
	dewaxed	120	114	95				

## Data Availability

Data are available upon request from the authors.

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
