# Peer review of "The Impact of Tissue Preparation on Salivary Gland Tumors Investigated by Fourier-Transform Infrared Microspectroscopy"

_jcm, 2023, doi:10.3390/jcm12020569_

Round 1

Reviewer 1 Report

To the Authors:

The manuscript represents a study of an infrared based tissue preparation in a limited salivary gland tumor subtype.

The following are the main issues:

1.      The rationale for utilizing this tissue preparation selectively in salivary gland tumor digitalization is unconvincing.  Except for applying this technique to a rare tumors, the concept is no difference from any tumors of different organs.

2.      Evidence that this technique is superior than traditional methods is lacking. In other words, how using it in salivary gland tumors different from any other tumors for diagnosis on H&E or biomarkers.

3.      The study is limited to pleomorphic adenoma and worthies’ tumor both are benign and molecular analysis are not needed for routine diagnostic purposes.

4.      No comparative biomarker analysis on different tissue preparations are done to demonstrate the superiority of this method.

Author Response

Dear Reviewer,

Thank you very much for your time and effort. We appreciate your valuable suggestions to improve the quality of our contribution. You can find the changes in the attached file.

Mona Stefanakis

Reviewer 2 Report

Very well written manuscript about the bio informatics to diagnostics of salivary gland tumor assement at molecular level assessment but the relevance of this method towards warthin salivary gland tumor homogenesity and sensitivity  being rated high versus pleomorphic adenoma should have explained, is this a draw back to the present study protocol, can paraffin embedded previous stock saved blocks be utilised for the molecular assessment using FTIR could have been explained, the advantages versus disadvantages of the present protocol in comparison towards existing techniques would have added more significance to the study stressing the importance of molecular bond of tissue and fat which later may be adapted for image analysis using mri, nevertheless a good study.

Author Response

(The authors gave the same response as above.)

Reviewer 3 Report

The introduction precisely explains the bology and histopathology of the two salivary tumors and the method of histopathological examination using spectroscopy of four different material samples.

Material and method properly designed and clearly presented results. Unfortunately, the results are not fully explained in the conclusion. we do not receive unequivocal conclusions and the authors state the need for further research.

Author Response

(The authors gave the same response as above.)
